# Community perspectives on maternal dietary diversity in rural Kenya, Mozambique and The Gambia: A PRECISE Network qualitative study

Mai-Lei Woo Kinshella[1], Shilla Dama[2], Onesmus Wanje[2], Rosa Pires[3], Helena Boene[3], Papa Jagne[4], Hawanatu Jah[4], Angela Koech[2], Grace Mwashigadi[2], Violet Naanyu[2,5], Yahaya Idris[4], Fatoumata Kongira[4], Brahima A. Diallo[4], Omar Ceesay[4], Marie-Laure Volvert[6], Hiten D. Mistry[6], Marleen Temmerman[2], Esperança Sevene[3,7], Anna Roca[4], Umberto D'Alessandro[4], Marianne Vidler[1], Laura A. Magee[1,6], Peter von Dadelszen[1,6], Sophie E. Moore[4,6], Rajavel Elango[8], the PRECISE Network[¶]

1 Department of Obstetrics and Gynaecology, BC Children's and Women's Hospital and University of British Columbia, Vancouver, Canada, 2 Centre of Excellence for Women and Child Health, Aga Khan University, Nairobi, Kenya, 3 Centro de Investigação em Saúde de Manhiça (CISM), Manhiça, Mozambique, 4 MRC Unit The Gambia at the London School of Hygiene and Tropical Medicine, Fajara, The Gambia, 5 School of Arts and Sciences, Moi University, Eldoret, Kenya, 6 Department of Women and Children's Health, King's College London, London, United Kingdom, 7 Faculdade de Medicina, Universidade Eduardo Mondlane, Maputo, Mozambique, 8 Department of Pediatrics, School of Population and Public Health, BC Children's and Women's Hospital and University of British Columbia, Vancouver, Canada

¶ Membership of The PRECISE Network is provided in Supporting Information file S1 Table.
* Maggie.kinshella@cw.bc.ca

## Abstract

Pregnant and lactating women in sub-Saharan Africa are vulnerable to micronutrient inadequacies, with risk of adverse pregnancy outcomes. Adequate intakes of diverse foods are associated with better micronutrient status and recommended by the World Health Organization as part of healthy eating counselling during antenatal care. However, our understanding of community knowledge of dietary diversity within the context of maternal diets is limited. We used a descriptive qualitative approach to explore community perceptions of dietary diversity during pregnancy and lactation, as well as influencing factors in sub-Saharan Africa. A total of 47 in-depth interviews were conducted between May and October 2022 in Kenya, Mozambique and The Gambia with a purposively drawn sample of pregnant women and mothers who had delivered within two years preceding the data collection, their male and female relatives, and community opinion leaders. Other methods included participant observation and photovoice. Data were analyzed using a thematic approach on NVivo software. Dietary diversity was found to be well aligned with local perceptions of healthy meals. All participants were able to differentiate between starchy staple grains and additional foods to provide nutrients. While diverse meals were valued for pregnant and lactating mothers, participants across the three countries shared that maternal diets were not more diverse compared to typical household meals. Furthermore, diverse diets were inaccessible for many in their communities, due to challenges in affordability, seasonality, gender norms, knowledge and preferences. Adequate nutrition

**Data availability statement:** All relevant data are within the paper and its Supporting Information files.

**Funding:** The PRECISE and PRECISE-DYAD studies are funded by UK Research and Innovation Grand Challenges Research Fund GROW Award scheme [MR/P027938/1 to PvD] and NIHR–Wellcome Partnership for Global Health Research Collaborative Award [217123/Z/19/Z to PvD]. The Canadian Institutes of Health Research (CIHR) provided funding in the form of the Vanier Canada Graduate Scholarship [to M-LWK]. The funders had no role in study design, data collection and analysis, decision to publish, or preparation of the manuscript.

**Competing interests:** The authors have declared that no competing interests exist.

knowledge, accessibility of foods, and support of household decision-makers, particularly husbands and partners, were all identified as critical to ensure women have adequate diverse maternal diets.

## Introduction

In 2022, an estimated 23% of the sub-Saharan African population faced hunger [1]. Dietary diversity is a key component of a quality diet; a varied diet increases the probability of nutrient adequacy and higher dietary diversity is associated with a decreased risk of all-cause mortality [2,3]. Low-diversity dietary patterns with insufficient intake of whole grains, fruits and vegetables have emerged as a leading contributor to morbidity and mortality globally [4]. Due to high nutritional requirements to optimize pregnancy outcomes, pregnant and lactating women are especially vulnerable to nutrient inadequacies [5–10]. However, dietary patterns of pregnant women in low- and middle-income countries (LMICs) are frequently dominated by starchy foods and characterized by inadequate micronutrient intake [11]. Substantial deficiency rates of iron, vitamin A, iodine, zinc and folate among pregnant and non-pregnant women of reproductive age (WRA) have been identified across four countries in sub-Saharan Africa, including up to 61% of pregnant women deficient in iron and up to 76% deficient in zinc [12]. Substantial gaps in meeting minimum dietary diversity for women have been reported in sub-Saharan Africa, including 31% of women with inadequate dietary diversity in Kenya and 48% in Mozambique between 2021–2022 [13]. Low maternal dietary diversity during pregnancy has been associated with a number of adverse pregnancy outcomes such as increased risk of maternal anemia, preterm birth, and delivering small-for-gestational age or low birthweight infants [14–18], as well as with increased rates of maternal depression and anxiety [19].

Although dietary diversity is recommended by the World Health Organization (WHO), to be part of healthy eating counselling during antenatal care for all women [20], the literature on community knowledge and perceptions of dietary diversity is sparse. Limited previous qualitative studies in Tanzania [21,22], Uganda [23], and Nigeria [24] have often focused on agricultural and economic factors and a gap remains on understanding local conceptualizations of dietary diversity within the context of maternal diets. This study explores community perceptions of dietary patterns and diversity during pregnancy, and their influencing factors among rural communities in Kenya, Mozambique and The Gambia.

## Methods

### Study design and research setting

We conducted a descriptive community-based qualitative study to investigate local perspectives and experiences of dietary diversity and healthy maternal diets in sub-Saharan Africa. Maternal diets were conceptualized as the food eaten during pregnancy and lactation. We employed a focused ethnographic approach, an applied exploratory methodology emphasizing rich descriptions focused on an area of interest, in contrast to a comprehensive description of a culture in traditional ethnographic methods [25–27]. A focused ethnographic approach also frequently features triangulation between a combination of qualitative methodologies to understand participant experiences [25–27]. Our research featured in-depth interviews (IDIs), which was triangulated with participant observation and photovoice to strengthen qualitative findings. Study design was developed and refined by an international team of qualitative experts (MWK, SD, MV, VN, BAD), clinician scientists (AK, HJ, IY, LAM,

PvD), PRECISE project coordinators from each country (GM, OW, FK, HB), and maternal and child nutrition experts (SEM, RE). Engagement of researchers, coordinators, and clinicians from Kenya, Mozambique and The Gambia helped to ensure cultural appropriateness and contextualization of methods and interpretation of findings relevant to each study community. The study is reported following the "Consolidated criteria for reporting qualitative research" (COREQ) [28].

This study is part of the larger PRECISE (PREgnancy Care Integrating translational Science, Everywhere) Network, a prospective pregnancy cohort in Kenya, The Gambia and Mozambique to investigate placental disorders and other complications of pregnancy in sub-Saharan Africa; qualitative research took place in PRECISE study communities to understand local contexts and social determinants of health [29,30]. These included Mariakani and Rabai, located in Kaloleni and Rabai sub-counties in Kilifi County in the coastal region of Kenya, Illiasa District in Kerewan (formerly the North Bank) of The Gambia, and Xinavane Administrative Post in Manhiça District in the Southern region of Mozambique. These communities were predominantly rural; only Mariakani was peri-urban.

The qualitative study received ethics approval from the University of British Columbia (H20-00143), the Aga Khan University Nairobi (2021/IERC-159), The Gambia Government/Medical Research Council The Gambia (MRCG) Joint Ethics Committee (26342), and the Centro de Investigação em Saúde de Manhiça (CISM) Institutional Review Board (CIBS-CISM/034/2022).

## Recruitment and selection

The study purposively drew a sample of pregnant women, recent mothers who had delivered within the previous two years, male and female family members from participating mothers' households, and community leaders, covering a diversity of age groups. As household decision-makers, mothers-in-law were prioritized for female relatives, and spouses (husbands and cohabiting/common-law partners) for male relatives, where possible. Community leaders included village elders, religious leaders, chiefs/assistant chiefs, traditional birth attendants/community birth companions, and others considered as influential opinion leaders in the community. Participants were required to be residents in the study communities.

A PRECISE Network study team member approached potential participants in person at the household or by phone to discuss the maternal diets qualitative sub-study. A sample size of 12-18 interviews in each country was estimated as required to reach data saturation with a variety of perspectives [31]. An additional three pregnant or recent mothers were recruited in each country for participant observation and five-to-ten pregnant or recent mothers for photovoice to confirm data saturation [32]. We oversampled pregnant and recent mothers to ensure inclusion of a diversity of perspectives. Literature has shown that young women's voices can be marginalized in patriarchal societies particularly where married couples reside in extended households such as traditionally practiced within the communities in this study [33].

## Data collection

Participant recruitment and data collection took place from May 5th to 31st, 2022 in Kenya, June 6th to 30th, 2022 in The Gambia, and September 12th to October 15th, 2022 in Mozambique. Local data collectors from Kenyan, Mozambican and Gambian PRECISE field teams (five females, six males) underwent an intensive training on qualitative methods and data collection materials prior to conducting interviews. Data collectors were fluent in local and national languages, which included Mijikenda and Swahili in Kenya, Mandinka, Fula, and

Wolof in The Gambia, Changana and Portuguese in Mozambique, and English across the three countries, and did not have any personal relationships with potential participants prior to the study. Communication between data collectors and study participants occurred in a language of comfort and good understanding by the participant. Across the three countries, potential participants were briefed in-person about the project by data collectors and informed consent obtained for each qualitative method prior to research activities.

Following a semi-structured topic guide piloted with local PRECISE staff (S1 File), data collectors conducted 45-to 120-minute face-to-face interviews at a location of participants' preference at a secluded place within the participant's household or the health facility. Interviews were conducted in the participants' preferred language. For participant observation, researchers (MWK [all countries], SD [Kenya], PJ [The Gambia], RP [Mozambique]) spent one to two full days to shadow each woman during her daily food provisioning activities in her household and community. Researchers participated in the process of obtaining food, such as from a store, garden or local market, then cooking and eating together. Fieldnotes were compiled during and after each participant observation session documenting informal conversations during observations on food provisioning, food preparation, meals, mealtime configurations within a family setting and overall conversations about maternal diets. Photovoice was conducted with an additional 20 women (five each in Mariakani and Rabai (Kenya); five in Illiasa (The Gambia); five in Xinavane (Mozambique). Digital cameras were distributed to women with training on how to use the cameras and take photographs, whilst maintaining privacy and participant safety. Cameras were collected by researchers approximately four days later. Photographs were used as prompts during in-person discussions either with each woman individually (The Gambia) or in a 90-minute focus group discussion (Kenya and Mozambique), as directed by local staff to be culturally appropriate in each country. Overall, there were no refusals to participate, and no repeat interviews were conducted. Debriefs with the data collectors during the data collection period ensured data quality and consistency.

## Data analysis

We employed a thematic approach in analyzing the data [34,35]. Fieldnotes and audios were recorded with permission. These were transcribed verbatim in Swahili (Kenya) and Portuguese (Mozambique), then translated into English. Audios from The Gambia were translated into English and transcribed simultaneously. Transcripts were uploaded to NVivo software for analysis. Participant characteristics were compiled from transcripts based on open-answered demographics questions in the interview guide. Confidentiality was ensured by de-identifying participants using codes and aggregating demographic features.

A coding framework was developed inductively from the data and deductively from research objectives to cover maternal dietary patterns and dietary diversity, community knowledge of healthy maternal diets, and food security (S2 Table). The coding framework was piloted by two independent researchers (MWK, SD), who double coded six interviews to refine the coding framework and ensure inter-rater reliability. The 41 remaining interviews were coded by MWK.

Themes from interview transcripts were enumerated by coding frequency overall (S3 Table) and by country and participant group (S4 Table). Following a thematic analysis approach, frequency of occurrence in transcripts was not used to determine the themes themselves, but rather to explore patterns of meaning from participants' perspectives [36]. In other words, themes that were more frequently mentioned across participant groups and countries may be considered more salient in participants' lived experiences; least frequently mentioned themes were reported as minor themes. Themes elicited from interview participants were triangulated with fieldnotes from participant observation and photovoice transcripts to support

credibility of findings [37,38]. Trustworthiness of key themes was reinforced by coverage in all methodologies, which examined maternal diets from a variety of perspectives.

A tree map was used to visualize patterns of dietary diversity influencing factors through illustrating the frequency of each theme relative to each other, with larger rectangles in the figure representing more frequent occurrence (based on numbers from S3 Table). In addition, a thematic map was constructed to illustrate the relationships between themes and summarize overall community perspectives regarding dietary diversity [36].

## Results

### Participant characteristics

There were 47 interviews, 18 from Kenya, 14 from Mozambique, and 15 from The Gambia (Table 1). Overall, these included 17 pregnant or recent mothers, 10 male relatives (spouses and bothers-in-law), nine female relatives (mothers-in law and sisters-in law) and 11 community leaders (traditional birth attendants, religious leaders, village elders and chiefs, local government officials). Among interview participants, 68% were female, 19% were 25 years old or younger, 32% were 26 to 35 years old, 15% were 36 to 45 years old, and 30% were 46 years or older. A minority (4%) did not know their age. Additionally, 94% reported that they were married or cohabitating with their partner, and 51% had two-to-four children. Two thirds of participants (64%) had some primary school attendance or no formal education and the most frequently reported occupation was farming (30%) followed by having a small business 21%. Other employment included one participant who reported receiving a small stipend from community volunteer work and another participant reporting a similar income for religious duties. Participants from The Gambia were largely reliant on farming, reporting that either they farmed or that they depended on the income of a family member who farmed. Participants from Kenya least frequently reported farming as their primary occupation and a third (33%) reported engaging in small business.

### Perceptions of dietary diversity

Overall, participants reported familiarity with dietary diversity, which was conceptualized as mixing different types of food either within a meal or between meals. Participants across the three countries and participant groups shared that consuming a variety of foods was important for health, including for pregnant and lactating mothers (S3 Table, S4 Table). Unmarried men and older mothers-in-law from The Gambia commented less on maternal diets, although they voiced the value of diverse diets for all people.

Participants reported a starchy base, typically comprised of grains as fundamental to a meal. This was mainly a stiff porridge made of maize meal in Kenya and Mozambique called *ugali/sima* or *xima,* respectively, and "cous" grains comprised mainly of millet or sorghum in The Gambia. In addition, rice was frequently consumed across all three countries and was reported by participants to be highly preferred.

While the participants felt that consuming the starchy base meal was filling, additional ingredients cooked as a stew served on top or cooked with the grains were required to add nutrition, with the generalization that more ingredients equated to healthier meals. Healthy meals were conceptualized to be "complete" with the inclusion of multiple food groups, particularly both meat/fish/beans and green leafy vegetables as well as a piece of fruit accompanying the staple grains in Kenya and Mozambique. Participants from The Gambia described multiple "condiments" such as fish, chicken, garden vegetables such as eggplants, bitter tomatoes, cabbage and onions cooked in the stew as adding to the nutrition and taste of food.

**Table 1. Demographic characteristics of interview participants.**

| | Kenya (n = 18) | Mozambique (n = 14) | The Gambia (n = 15) | Total (n = 47) |
|---|---|---|---|---|
| **Participant group** | | | | |
| Pregnant or mother | 7 (39%) | 5 (36%) | 5 (33%) | 17 (36%) |
| Spouse or brother-in-law | 4 (22%) | 3 (21%) | 3 (20%) | 10 (21%) |
| Mother-in-law or sister-in-law | 3 (17%) | 3 (21%) | 3 (20%) | 9 (19%) |
| Community leader | 4 (22%) | 3 (21%) | 4 (27%) | 11 (24%) |
| **Sex** | | | | |
| Female | 13 (72%) | 9 (64%) | 10 (67%) | 32 (68%) |
| Male | 5 (28%) | 5 (36%) | 5 (33%) | 15 (32%) |
| **Age** | | | | |
| 25 years old or younger | 6 (33%) | 1 (7%) | 2 (13%) | 9 (19%) |
| 26 to 35 years old | 3 (17%) | 6 (43%) | 6 (40%) | 15 (32%) |
| 36 to 45 years old | 4 (22%) | 2 (14%) | 1 (7%) | 7 (15%) |
| 46 years or older | 4 (22%) | 5 (36%) | 5 (33%) | 14 (30%) |
| Unknown | 1 (6%) | 0 (0%) | 1 (7%) | 2 (4%) |
| **Marital status** | | | | |
| Married or cohabitating | 18 (100%) | 14 (100%) | 12 (80%) | 44 (94%) |
| Single | 0 (0%) | 0 (0%) | 2 (13%) | 2 (4%) |
| Widowed | 0 (0%) | 0 (0%) | 1 (7%) | 1 (2%) |
| **Number of children** | | | | |
| 0 to 1 child | 4 (22%) | 1 (7%) | 2 (13%) | 7 (15%) |
| 2 to 4 children | 11 (61%) | 9 (64%) | 4 (27%) | 24 (51%) |
| 5 or more children | 3 (17%) | 4 (29%) | 9 (60%) | 16 (34%) |
| **Highest level of education attained** | | | | |
| No formal education | 3 (17%) | 1 (7%) | 13 (87%) | 17 (36%) |
| Some primary school | 6 (33%) | 5 (36%) | 2 (13%) | 13 (28%) |
| Completed primary school | 4 (22%) | 8 (57%) | 0 (0%) | 12 (26%) |
| Completed secondary school | 1 (6%) | 0 (0%) | 0 (0%) | 1 (2%) |
| Post-secondary education | 4 (22%) | 0 (0%) | 0 (0%) | 4 (9%) |
| **Occupation** | | | | |
| Farming | 1 (6%) | 6 (43%) | 7 (47%) | 14 (30%) |
| Small business | 6 (33%) | 3 (21%) | 1 (7%) | 10 (21%) |
| Labourer, house help or informal work | 3 (17%) | 4 (29%) | 0 (0%) | 7 (15%) |
| Professional or government employment | 2 (11%) | 0 (0%) | 0 (0%) | 2 (4%) |
| Housewife | 5 (28%) | 0 (0%) | 3 (20%) | 8 (17%) |
| Dependent on adult children | 0 (0%) | 0 (0%) | 4 (27%) | 4 (9%) |
| Other | 1 (6%) | 1 (7%) | 0 (0%) | 2 (4%) |

"Meals become healthy when you mix different nutrients in it. Just like today, you would eat green vegetables, which have vitamins. Tomorrow you get some meat, which is protein, or mix the green vegetables with meat, boil and give that to the pregnant mother, then get some *ugali* (stiff maize meal porridge), which is carbohydrates. That is a balanced diet and that is what makes a meal complete." *Spouse, Kenya*

"Foods now are more delicious now…because they have more ingredients than that of the past… Pregnant women should eat things like meat, leafy greens, vegetables, and seafood like fish to help them become healthy…If pregnant women fail to get enough of these

varieties of foods… they develop a lot of complications such as anemia etc." *Community birth companion, The Gambia*

"It's eating different things with vitamins… As soon as I am pregnant, I want fried *carapau* (horse mackerel fish) with onion rice, I want roast chicken… I have to eat fruit. I have to eat things from the farm (vegetables) to increase blood for the child to grow well" *Pregnant woman, Mozambique*

Participants emphasized that additional food groups provided "vitamins," which were seen to be especially important during pregnancy and lactation to support the growing baby and prevent complications such as anemia. Nutrients were generalized broadly as "vitamins"; specific vitamins nor minerals were not mentioned. In contrast, a lack of dietary diversity, described as eating only the staple foods or the same foods for multiple meals, was considered unhealthy for pregnant women.

"If you are pregnant and you wake up early in the morning to eat leftover food from yesterday…dried fish with cold *ugali*, that is not good nutritious food and it can even kill you… Also… eating ugali with salt…is not allowed" *Pregnant woman, Kenya*

"If they want to depend on only rice, it sometimes gives the problems…. For rice, people like it but is not a health-giving food. Just things that are put on top of the rice, which give vitamins, but not the rice… Types of food like "*Nyankatang*"….has multiple condiments… it gives a lot of vitamins to a pregnant woman." *Community leader, The Gambia*

"The meat itself, but… don't miss small vegetables…don't miss an orange and banana… Together they make a healthy diet…. When they [pregnant women] consume meat with *xima* only…The meal is decent, but not nutritious" *Community leader, Mozambique*

Interview results on perceptions of healthy diverse maternal diets were reinforced by similar findings from photovoice, as illustrated in Fig 1. Through the photographs women took, they described the value of balanced meals with multiple food groups in Kenya, varying meals in Mozambique, and health benefits of dishes with more ingredients in The Gambia.

## Perceptions of influencing factors

While dietary diversity was described as ideal for health, participants across the three countries described a limited number of foods within typical household meals and that diverse diets were not achievable for most in their communities. Further, it was reported that maternal diets did not substantially differ from typical household meals. Affordability and seasonality were described as leading factors that influenced the types of food pregnant and lactating mothers consumed in their communities, followed by knowledge, gender norms and cravings (Fig 2). Traditional beliefs, religion, and rural residency were least frequently mentioned factors. Participants from all three countries (S3 Table) and across all three participant groups (S4 Table) highlighted the same factors, though the emphasis varied between countries and groups. The consequence of irregular weather patterns, particularly drought due to lack of rainfall, was especially a concern voiced by Kenyan participants. Gambian participants more frequently described a lack of certain foods in rural communities and less frequently reported cravings and traditional beliefs as drivers for the types of food pregnant women eat. The influence of pregnancy food cravings on maternal diets was especially emphasized in Mozambique. Illustrative quotes across countries can be found in S5 Table.

| Healthy foods for maternal diets described by participants | Typical meals consumed by women and their families |
|---|---|

**Kenya**

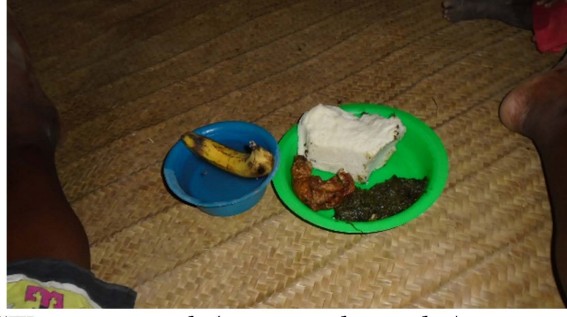

*"We can see ugali (maize meal porridge), green vegetables, meat and a banana… That photo is very important. Because when a woman is pregnant and gets such kind of food, it makes it easy for her to gain weight."*

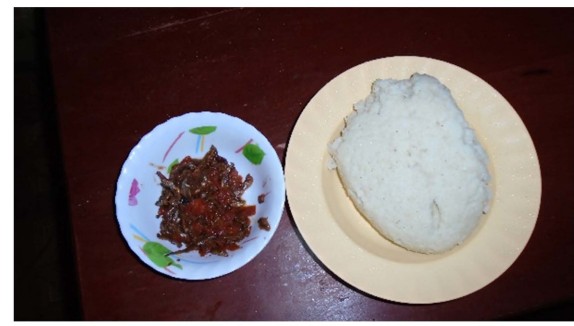

*"This picture shows ugali (maize meal porridge) and omena (small dried fish cooked in oil and tomatoes). This means that you cannot have ugali only, you need mboga (vegetables, including fish) and vice versa. Hence, one needs to work hard to get both."*

**Mozambique**

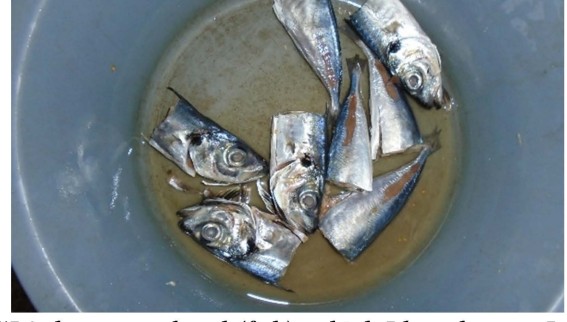

*"It's horse mackerel (fish), which I bought… so I can cook it… [I]deal foods for us to eat… [is] vegetables, fish, [when] we're varying the foods.*

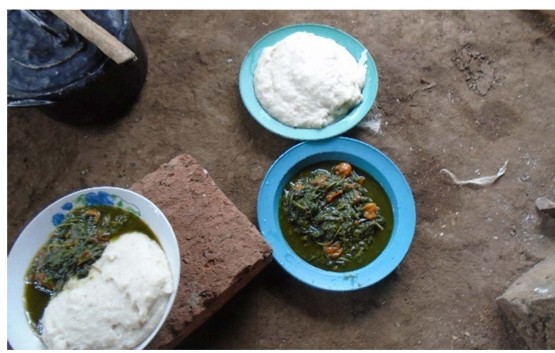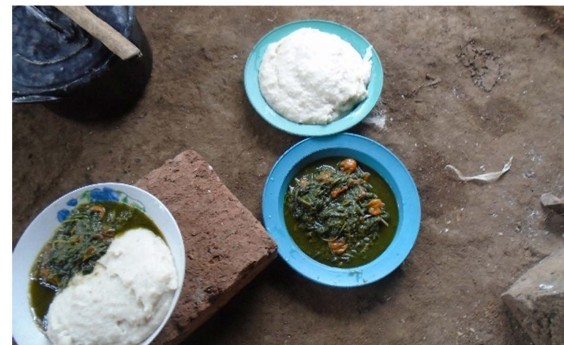

*"Here I was still pregnant! It's madledlele (sweet potato leaves), put rhulana (cherry tomato)! …Eat food like vegetables so they (pregnant women) can have vitamins!"*

**The Gambia**

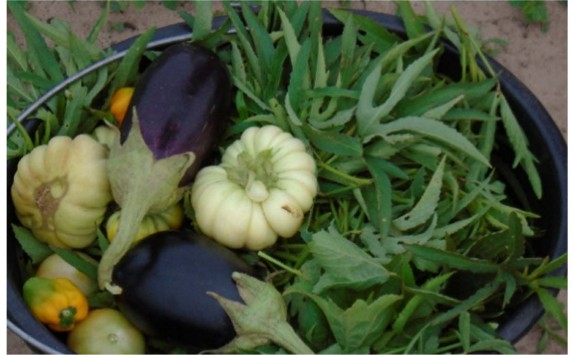

*"Sorrel leaves, garden egg (eggplant) and bitter tomato… these are very good for our body…so that you don't have problems with your blood and they also help strengthen your bodies immunity. That's why we cultivate them"*

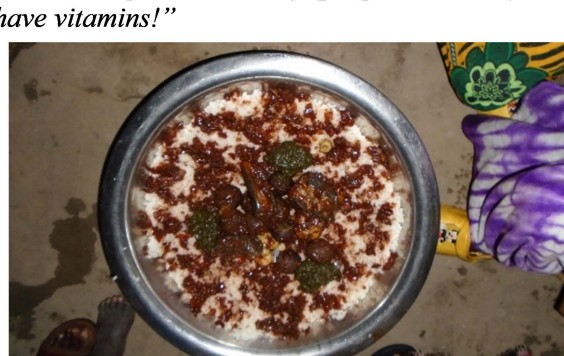

*"This is bullet fish in oil sauce with pounded sorrel and other vegetables. We usually cook this type of food because they are very useful to human health"*

**Fig 1. Examples of maternal diets described by women during photovoice.**

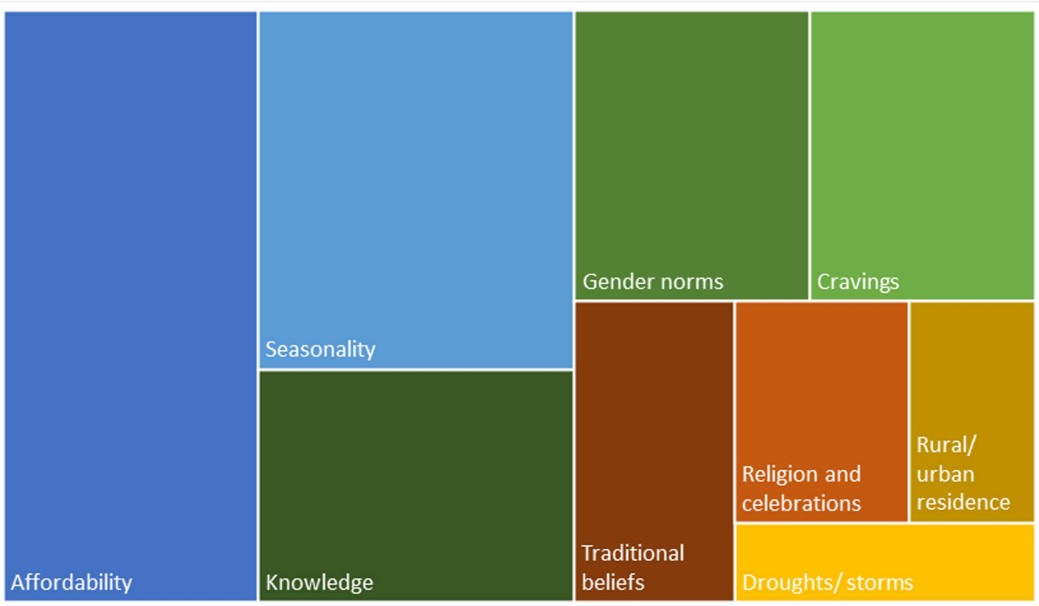

**Fig 2. Frequency map of reported themes for dietary diversity influencing factors.**

Affordability and seasonality, the most frequently mentioned factors, were especially challenging for pregnant and lactating women as participants shared that nutritious foods were often purchased or seasonal. Fruits and vegetables were more available during the rainy season in each respective country, although vegetables were grown during the dry season in The Gambia by those with gardens. Kenyan participants emphasized the ongoing drought in their region as a driving factor for limiting the types of food consumed, highlighting crop failures of fruits, vegetables and legumes and reduced grazing for local cows to produce milk due to unreliable and reduced rains.

"There are specific types of food that pregnant women and lactating mothers should depend on but due to lack of affordability, we all depend entirely on rice and coos (millet)…" *Spouse, The Gambia*

"In summer, we don't have vegetables… The months of April, May and June, those are the winter period…. Until July, we still have something. In August, there is nothing in the farms" *Mother-in-law, Mozambique*

"The problem is like now when there is drought. There is no rain at all there is no food in the *shambas* (farms) so you cannot get healthy foods." *Spouse, Kenya*

Participants described seasonality as interconnected with affordability because out-of-season crops required more money to purchase. In comparison to the past, Gambian participants reported increasing accessibility to different types of food in part due to monetary remittances from family members working abroad. However, participants from Kenya and Mozambique described substantial financial barriers to dietary diversity.

"Kale is also eaten. It's just expensive… [when] the person can no longer plant because it is already very hot, and when they buy…sometimes…there is no money to buy anything else..." *Pregnant woman, Mozambique*

"[We are] helped by some of our children living abroad with some money to buy food stuff….It was very difficult in the past… because of poverty…. Now things have changed, you talk of affordability but not availability; you can buy anything you want if you have money" *Community leader, The Gambia*

"It is too expensive and very difficult to afford change of food aside from *ugali*. We just eat *ugali* in the afternoon and in the evening…Affordability, affordability. I think you can see that things are not that easy. People do not have jobs, they are not employed and do not have even casual jobs. People are burdened." *Community leader, Kenya*

Other major influencing factors described by participants included knowledge of healthy foods during pregnancy and lactation, local gender norms about household decision-makers, and pregnancy food cravings. Clinics were described as a key source of information on healthy maternal diets, but many participants shared that their knowledge was largely from experience, either from their own pregnancies or from family members. In all three countries, men were conceptualized as the primary provider for the household and the types of food consumed by pregnant women relied on what their spouses brought home or provided funds for. Emphasizing cravings was a way for women to request certain foods during pregnancies, although participants sometimes discounted the food women wanted as unnecessary cravings.

"Whenever you go to the hospital, the doctor advises to eat fruits, vegetables and soup this helps a lot with increasing blood and energy" *Recent mother, Kenya*

"If available, they [women] eat and if not, they just sit and wait… That is it… It is your husband who feeds and protects you." *Traditional birth attendant, The Gambia*

"I have a daughter who during her pregnancy she didn't eat anything other than oranges… That's because it's not the woman's will…But…what she carries in her womb." *Community leader, Mozambique*

Factors less frequently mentioned by participants included traditional beliefs about pregnancy food taboos, religion and a rural/urban divide. There were some reports about food that should be avoided during pregnancy, including eggs, bread, meat and seafood. However, participants shared that these beliefs were changing with increasing education about healthy foods during pregnancy and witnessing positive pregnancy outcomes among women who consumed these healthy foods in their communities. Mentions of religious food restrictions were rare and participants only described pork as prohibited among Muslims. Participants more frequently mentioned religious holidays and other major celebrations as occasions where meat and chicken were consumed. Some food items were also difficult to find in rural and remote communities. Imported and out-of-season fruits, and meat were reported to be more widely available in urban and peri-urban areas.

"What they said in the old days was myth… she can eat eggs…because it provides vitamins for the baby." *Mother-in-law, Mozambique*

"We eat meat during Eid because my husband's family is Muslim and they celebrate Eid. For me, because I am Christian, I can cook chicken for Christmas." *Pregnant woman, Kenya*

"We do not have oranges here [out of season]. If you go to the urban areas and you see it there, you [can] buy… Pregnant women, when they go for the antenatal clinics and see it being sold, they buy and eat." *Community leader, The Gambia*

## Thematic map

Fig 3 illustrates the community perspectives and experiences of dietary diversity described by study participants, nested within local conceptualizations of meals. Dietary diversity influencing factors were sometimes described as interconnected, such as affordability, seasonality and irregular weather patterns, as well as women's food cravings negotiated through gender norms, the knowledge men had about healthy foods during pregnancy, and changing traditional beliefs with health education from clinics.

# Discussion

## Summary of findings in comparison to the literature

This research in three sub-Saharan African countries found that dietary diversity was well-aligned with community members' understanding of healthy meals. Dietary diversity was seen to be particularly valuable for pregnant and lactating mothers to prevent anemia and support

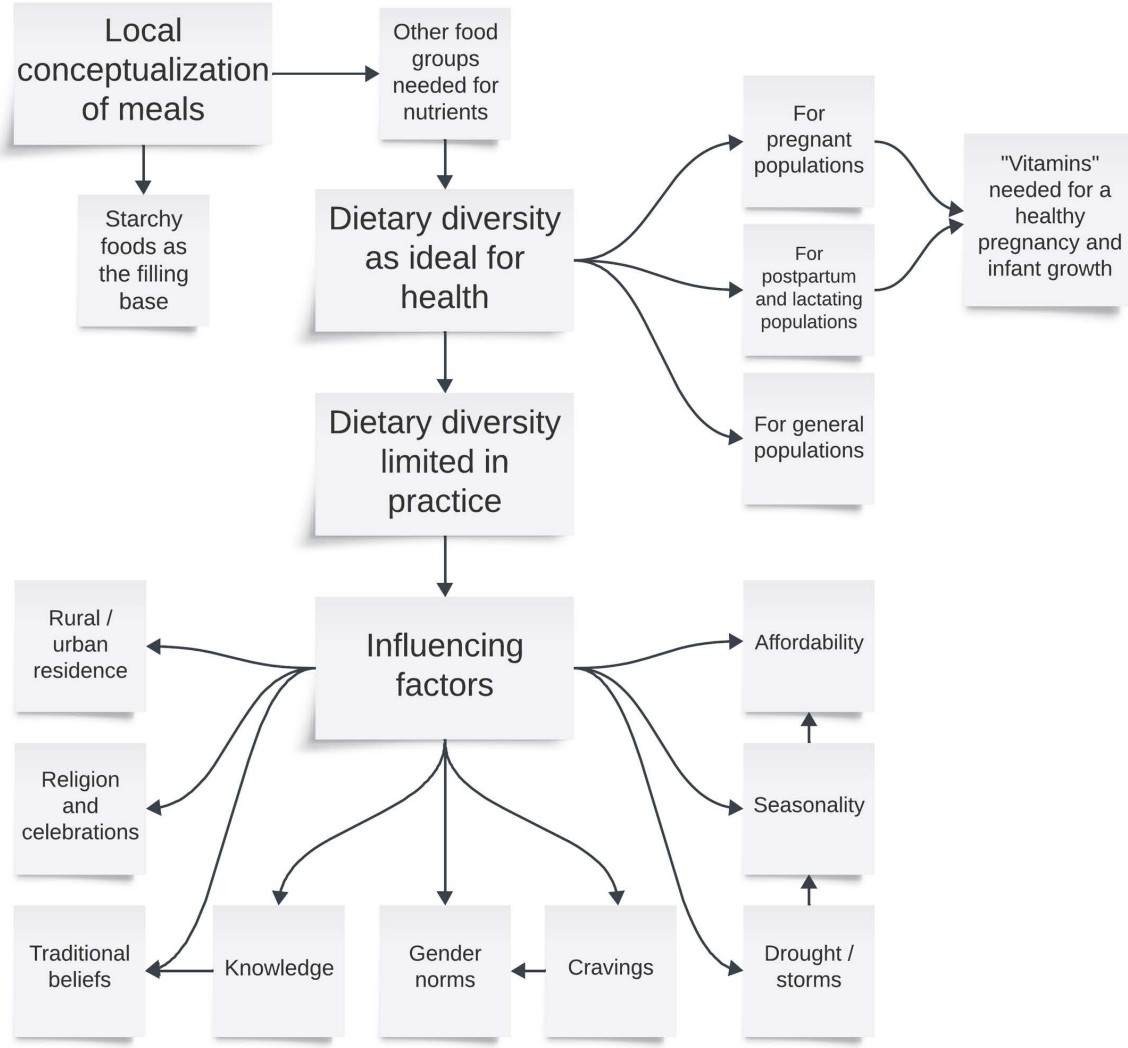

**Fig 3. Thematic map of community perspectives on dietary diversity among maternal and general populations.**

healthy infant growth. In contrast to a previous study among farming households where dietary diversity was seen largely to promote the appeal of meals and increase appetite rather than for nutritive value [21], participants in this study frequently emphasized the importance of diverse diets to provide "vitamins," probably reflecting maternity and child health education. Consequently, there may be heightened awareness about the importance of dietary diversity, especially for pregnant and recent mothers, yet community members reflected on significant barriers. Participants indicated limited accessibility due to a complex interplay of affordability, seasonality, gender norms, local knowledge and preferences.

While dietary diversity was highly meaningful within local understandings of meals, hierarchies in the priority of various foods challenged the capacity for diverse maternal diets. Carbohydrate staples of lower nutritive value were prioritized as fundamental components of meals to feel full, in comparison to costly nutritious foods like fruit, eggs, meat and out-of-season vegetables. Affordability and seasonality were widely described as key challenges to the practice of dietary diverse meals, similar to Tanzania [21], Uganda [23], and Nigeria [24]. Participants in our study highlighted that affordability challenges limited the types of foods that households could access, rather than necessarily equated with hunger. This aligns with research reporting fewer types of foods consumed with increasing food insecurity [39,40] and that food insecurity did not equate to lower energy intakes [41]. Two large systematic reviews found that increasing food prices or falling income due to job instability created pressure to purchase foods that are lowest in cost, particularly in economically disadvantaged settings and incomes [42,43].

Participants in this study highlighted an increasing emphasis on cash, which may reflect demographic and nutritional transitions in rural sub-Saharan Africa. Previous studies emphasized agriculture as the mainstay economic activity for rural sub-Saharan African households [22] and highlighted diversity in household food production for dietary diversity [21,23]. However, less than a third of those interviewed here reported farming as their primary livelihood activity and rural residence was not described as a major influencing factor due to the growing availability of shops and vendors, though there were regional differences. Our study reflects emerging findings in the State of Food Security and Nutrition in the World 2023 report of increasing access to a variety of food (including processed foods) in rural and remote global settings and that even rural populations purchase a majority of their food [1].

While gender norms have been recognized to influence household nutrition and dietary diversity practices [23,44–47], a growing dependence on cash may shift household gender dynamics to further disadvantage diverse maternal diets. Rural women are traditionally in charge of food crops for household consumption, while men are more likely responsible for decision making regarding cash crops and other sources of income [22]. A study from Nigeria found that women often required permission before purchasing costly food items [24], which may include nutrient dense foods during pregnancy and lactation. Thus, an increased reliance on money may deepen dependency on men as primary household providers and decision-makers for food and other essentials.

## Implications for research and policy

Dietary diversity recommendations during pregnancy resonate with local knowledge about healthy eating, which supports nutrition counselling during antenatal care. However, this research highlighted the contribution of gender norms and family dynamics to maternal diets, with a complex interplay between the types of food women wanted during pregnancy and lactation, economic constraints, gender roles, and knowledge. These findings support the need for further research and policies to engage men in maternal nutrition education [48,49] and the economic empowerment of women [24,47,50–52]. In addition to gender norms around

food provisioning, further exploration on the intersections between maternal diets and gender norms around food preparation, household responsibilities, and meal consumption are recommended.

Additionally, local conceptualizations of more ingredients equating to healthier meals is complicated by the surge of processed foods in sub-Saharan Africa, even in rural communities [1,53–55]. There is a concern that higher dietary diversity may be associated with more frequent consumption of low nutrient density foods [56–59] and more research is needed to understand dietary diversity in sub-Saharan Africa with the local context to ensure optimal dietary counselling.

## Strengths and limitations

Strengths of this study include rigorous data collection procedures, reporting using a checklist to ensure dependability, use of clear coding framework, and the engagement of local and international field experts in analysis and interpretation to support confirmability. Our study may be limited by short period of observation, and social desirability bias in reporting, but these are in part mitigated by a diversity of participants recruited for interviews and the triangulation of findings between qualitative methods to strengthen multiple angles of perspectives. Additionally, qualitative research can be limited in generalizability beyond study communities, though research in three countries strengthens our capacity to explore overarching common themes and similarities across three district African geographies while highlighting areas where each location may be unique.

## Conclusion

Our research in Kenya, Mozambique and The Gambia found that dietary diversity was consistently considered by community members important for healthy maternal diet, which supports nutrition education and counselling during pregnancy as recommended by the WHO [20]. However, meals for pregnant and recent mothers were usually similar to their households and consisted of a limited number of foods. Moreover, there may be potential misconceptions around dietary diversity and the consumption of low nutrient density foods. Appropriate nutrition knowledge and accessibility are both needed to ensure women have adequate diverse maternal diets to support maternal and perinatal health. Exploring further the strong role of men as economic source and household decision-maker as to which foods are purchased alongside with their level of knowledge around dietary diversity and specific dietary needs of pregnant women and recent mothers may support enhancing the maternal dietary and health status. More research is needed on adequate maternal diets within the context of nutritional transitions.

## Supporting information

**S1 File. Semi-structured interview guide.**
(DOCX)

**S1 Table. The PRECISE Network.**
(DOCX)

**S2 Table. Qualitative analysis coding framework.**
(DOCX)

**S3 Table. Frequency of reported themes by country and overall.**
(DOCX)

**S4 Table. Frequency of themes reported in each participant group, by number of participants.**
(DOCX)

**S5 Table. Community perspectives on influencing factors by country.**
(DOCX)

**S1 Data. COREQ Checklist.**
(PDF)

**S1 Checklist. Inclusivity in global research.**
(DOCX)

## Acknowledgements

The authors would like to express their gratitude to the PRECISE Team for their support and all of the women and their families who participated in the study. This manuscript is part of the PRECISE (PREgnancy Care Integrating translational Science, Everywhere) Network.

## Author contributions

**Conceptualization:** Mai-Lei Woo Kinshella, Sophie E. Moore, Rajavel Elango.

**Data curation:** Shilla Dama, Helena Boene, Papa Jagne, Omar Ceesay.

**Formal analysis:** Mai-Lei Woo Kinshella, Shilla Dama, Onesmus Wanje, Helena Boene, Angela Koech, Grace Mwashigadi, Rajavel Elango.

**Funding acquisition:** Mai-Lei Woo Kinshella, Peter von Dadelszen.

**Investigation:** Mai-Lei Woo Kinshella, Shilla Dama, Onesmus Wanje, Rosa Pires, Helena Boene, Papa Jagne, Angela Koech, Fatoumata Kongira.

**Methodology:** Mai-Lei Woo Kinshella, Shilla Dama, Onesmus Wanje, Rosa Pires, Helena Boene, Hawanatu Jah, Angela Koech, Grace Mwashigadi, Violet Naanyu, Yahaya Idris, Fatoumata Kongira, Brahima A. Diallo, Omar Ceesay, Marleen Temmerman, Esperança Sevene, Anna Roca, Umberto D'Alessandro, Marianne Vidler, Laura A. Magee, Peter von Dadelszen, Sophie E. Moore, Rajavel Elango.

**Project administration:** Mai-Lei Woo Kinshella, Onesmus Wanje, Helena Boene, Hawanatu Jah, Angela Koech, Grace Mwashigadi, Yahaya Idris, Fatoumata Kongira, Omar Ceesay, Marie-Laure Volvert, Hiten D. Mistry.

**Supervision:** Mai-Lei Woo Kinshella, Shilla Dama, Onesmus Wanje, Helena Boene, Hawanatu Jah, Angela Koech, Omar Ceesay, Marie-Laure Volvert, Hiten D. Mistry, Marleen Temmerman, Esperança Sevene, Anna Roca, Umberto D'Alessandro, Marianne Vidler, Laura A. Magee, Peter von Dadelszen, Sophie E. Moore, Rajavel Elango.

**Visualization:** Mai-Lei Woo Kinshella.

**Writing – original draft:** Mai-Lei Woo Kinshella.

**Writing – review & editing:** Mai-Lei Woo Kinshella, Shilla Dama, Onesmus Wanje, Rosa Pires, Helena Boene, Papa Jagne, Hawanatu Jah, Angela Koech, Grace Mwashigadi, Violet Naanyu, Yahaya Idris, Fatoumata Kongira, Brahima A. Diallo, Omar Ceesay, Marie-Laure Volvert, Hiten D. Mistry, Marleen Temmerman, Esperança Sevene, Anna Roca, Umberto D'Alessandro, Marianne Vidler, Laura A. Magee, Peter von Dadelszen, Sophie E. Moore, Rajavel Elango.

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
