## [Decision Letter · Decision Letter 0]

10 Dec 2024

PGPH-D-24-01813

Community perspectives on maternal dietary diversity in rural Kenya, Mozambique and The Gambia: a PRECISE Network qualitative study

Dear Dr. Kinshella,

Thank you for submitting your manuscript to PLOS Global Public Health. After careful consideration, we feel that it has merit but does not fully meet PLOS Global Public Health’s publication criteria as it currently stands. Therefore, we invite you to submit a revised version of the manuscript that addresses the points raised during the review process.

We look forward to receiving your revised manuscript.

Kind regards,

Tsitsi B. Masvawure, Ph.D.

Academic Editor

Journal Requirements:

1. We do not publish any copyright or trademark symbols that usually accompany proprietary names, eg (R), (C), or TM  (e.g. next to drug or reagent names). Please remove all instances of trademark/copyright symbols throughout the text, including ® on page 26.

Additional Editor Comments (if provided):

This paper examines maternal dietary diversity in three countries: Kenya, The Gambia and Mozambique. The authors found that many community members understood the importance of dietary diversity for pregnant women but highlighted affordability and food insecurity due to climate change as reasons why pregnant women are not always able to eat healthy meals. Overall, the paper is well written and the research methods used appropriate. However, I would like to echo, and add to the reviewer's comments.

1. Study significance/"so what": the authors need to more strongly articulate the importance of the study and of their study findings. This can be done by revising the study goals to make them more compelling. Currently, the study goals seem to be aimed at presenting community perspectives on maternal dietary diversity but there is not much explanation of why. The introduction could do a better job at telling the reader what the maternal malnutrition or maternal food insecurity is globally and in the context of the three countries. Is data on this available? Furthermore, the authors should say why this topic is important: is lack of dietary diversity linked to adverse child health outcomes (e.g., low birth weight) or to poor maternal health outcomes (delivery complications etc)? Please expand your background section so it provides this critical context.

2. As per the reviewer's comments, can you present some of the photo voice data and some of the ethnographic data? This may help the reader gain a better sense of the actual diets consumed by pregnant women in the communities concerned.

3. I also invite you to consider the specific feedback provided by the reviewer.

Reviewers' comments:

Reviewer's Responses to Questions

**Comments to the Author**

1. Does this manuscript meet PLOS Global Public Health’s publication criteria ? Is the manuscript technically sound, and do the data support the conclusions? The manuscript must describe methodologically and ethically rigorous research with conclusions that are appropriately drawn based on the data presented.

Reviewer #1: Yes

2. Has the statistical analysis been performed appropriately and rigorously?

Reviewer #1: Yes

3. Have the authors made all data underlying the findings in their manuscript fully available (please refer to the Data Availability Statement at the start of the manuscript PDF file)?

Reviewer #1: No

4. Is the manuscript presented in an intelligible fashion and written in standard English?

Reviewer #1: Yes

5. Review Comments to the Author

Reviewer #1: 1. The research methods are robust and analysis is well conducted. Results of the Ethnographic research part of the study has has not been sufficiently described. Could photographic data collected in study area be presented to the readers?. Especially the data on usual types of food consumed and raw materials sold in market. Do markets sell mostly carbohydrate related food?

2. Pricing details of foods can be used to triangulate the findings in the affordability theme. Price inflation and Climate change impact can be mentioned in discussion. How does the community perceive affordability with regards to carbohydrates and other nutrients- do they feel hunger is to be satisfied first and then we can worry about the nutrients? More specifically as women- who are the usual cooks at home, what additional tasks, the women have to do to make the diet diverse, inside the kitchen. if this data is not available, authors can describe in study settings.

3. Socio-political milieu of the study settings in relation to dietary diversity such as Government schemes, Ethnicity and social hierarchies, civil conflicts, NGOs and projects related maternal nutrition needs mention

4. Was there any observed difference between primi and multigravida with regards to the study objectives?

5. Authors mention gender norms. However there is no description of gender norms around cooking, household tasks related to cooking, meal planning and dish washing associated with study objectives. Pregnancy craving is mentioned but was nausea mentioned as a barrier to eating healthy. Also is there a practice of men eating before women even during pregnancy? do children eat before women? are the best served for husband and children? Was this practice described by the participants? if this data is not available, authors can describe in study settings.

6. What is the public health implication of the conclusion of the study?

6. PLOS authors have the option to publish the peer review history of their article (what does this mean? ). If published, this will include your full peer review and any attached files.

**Do you want your identity to be public for this peer review?** For information about this choice, including consent withdrawal, please see our Privacy Policy .

Reviewer #1: **Yes: ** Nancy Angeline Gnanaselvam

---

## [Editor Report · Decision Letter 1]

26 Feb 2025

Community perspectives on maternal dietary diversity in rural Kenya, Mozambique and The Gambia: a PRECISE Network qualitative study

PGPH-D-24-01813R1

Dear Ms Kinshella,

We are pleased to inform you that your manuscript 'Community perspectives on maternal dietary diversity in rural Kenya, Mozambique and The Gambia: a PRECISE Network qualitative study' has been provisionally accepted for publication in PLOS Global Public Health.

Best regards,

Tsitsi B. Masvawure, Ph.D.

Academic Editor

Thank you for the revisions you made to your paper. The addition of the photo voice data is very helpful. I also appreciate your detailed responses to the reviewers.